# Broadly-Exploring, Local-Policy Trees
# for Long-Horizon Task Planning

**Brian Ichter**
Robotics at Google
ichter@google.com

**Pierre Sermanet**
Robotics at Google
sermanet@google.com

**Corey Lynch**
Robotics at Google
coreylynch@google.com

**Abstract:** Long-horizon planning in realistic environments requires the ability to reason over sequential tasks in high-dimensional state spaces with complex dynamics. Classical motion planning algorithms, such as rapidly-exploring random trees, are capable of efficiently exploring large state spaces and computing long-horizon, sequential plans. However, these algorithms are generally challenged with complex, stochastic, and high-dimensional state spaces as well as in the presence of small, topologically complex goal regions, which naturally emerge in tasks that interact with the environment. Machine learning offers a promising solution for its ability to learn general policies that can handle complex interactions and high-dimensional observations. However, these policies are generally limited in horizon length. Our approach, Broadly-Exploring, Local-policy Trees (BELT), merges these two approaches to leverage the strengths of both through a task-conditioned, model-based tree search. BELT uses an RRT-inspired tree search to efficiently explore the state space. Locally, the exploration is guided by a task-conditioned, learned policy capable of performing general short-horizon tasks. This task space can be quite general and abstract; its only requirements are to be sampleable and to well-cover the space of useful tasks. This search is aided by a task-conditioned model that temporally extends dynamics propagation to allow long-horizon search and sequential reasoning over tasks. BELT is demonstrated experimentally to be able to plan long-horizon, sequential trajectories with a goal conditioned policy and generate plans that are robust.

**Keywords:** RRT, Task and Motion Planning, Model-based Planning, Tree-search

## 1 Introduction

For a robot to plan complex long-horizon tasks in realistic environments, it must be capable of reasoning over sequences of subtasks. It must broadly search the space of possible trajectories while considering both what the subtasks are and the order in which they are accomplished. It must handle the complex and high dimensional state space and dynamics of the real world. Consider, for example, a robot cleaning a kitchen. A planner must find trajectories to put a utensil away, to wash a pot, to throw away trash, and more. Each of these tasks requires the ability to understand and execute lower-level tasks involved, as well as their order; to put a utensil away, the robot must first open a drawer, then grasp the utensil, then place it in the drawer, and finally close the drawer.

Classical planning algorithms are capable of efficiently and broadly searching state spaces and planning over long-horizons [1, 2]. However, they have difficulty in the presence of complex, stochastic dynamics, high-dimensional systems, so-called narrow passages, and small, complex goal regions (which tasks naturally create). Machine learning has emerged as the state of the art for problems with high-dimensional observations (e.g., pixels) and challenging dynamics (e.g., environment interaction). Furthermore, it allows learning policies for general and abstract tasks [3, 4, 5, 6, 7]. However, many such approaches are limited to short-horizons [8, 9].

We propose a method which leverages the best of both approaches, a tree search which explores the state space in an RRT-like manner, but where edges maintain a consistent task executed by a learned, task-conditioned policy (Fig. 1). In this work, we define task in a general sense, with a requirement only that the task space is sampleable and well-covers the space of useful actions; for example, this includes goal-conditioned policies, explicit tasks, learned skills, and latent task embeddings. The local, task-conditioned policy is thus capable of executing complex short-horizon tasks. The skeleton of the RRT search allows rapid exploration of the state space. This exploration is aided by a task-conditioned model, capable of rolling out long-horizon trajectories, which can then

5th Conference on Robot Learning (CoRL 2021), London, UK.

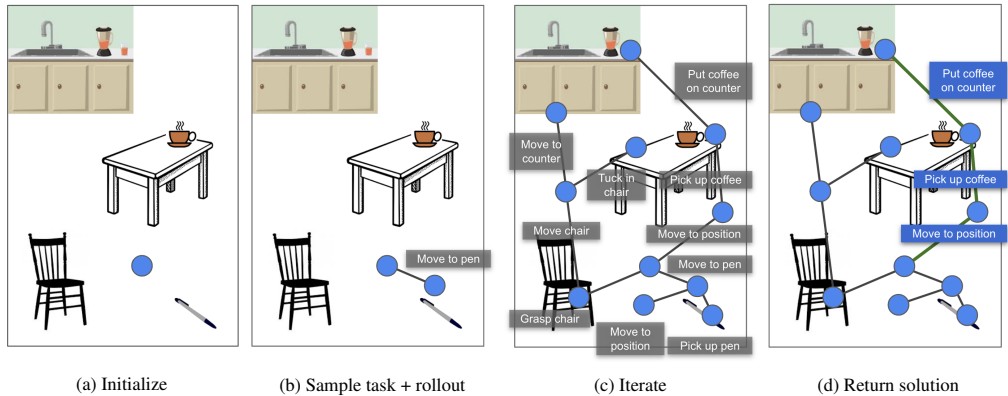

|(a) Initialize|(b) Sample task + rollout|(c) Iterate|(d) Return solution|

Figure 1: BELT plans long-horizon trajectories via a task-conditioned tree and task-model.

be planned over and reasoned about in a model-based manner to the limit of the known, predictable environment. Broadly-Exploring, Local-policy Trees (BELT) is thus able to create long-horizon plans with sequential, complex tasks.

## 1.1  Related Work.

*Planning Algorithms* are capable of efficiently, and globally, exploring state spaces to plan long-horizon trajectories which avoid collision [1, 2, 10]. Within robotics, as the robot must interact with the world, planning algorithms often consider the task and motion planning setting (contrasting with AI planning). Approaches in this setting generally plan in a hierarchical manner by defining *explicit* task-primitives, symbolic languages, or transition logic constraints. These methods then plan over them, for example, via trees or optimization-based methods [11, 12, 13, 14, 15, 16, 17]. These methods perform well in long-horizon planning, but are limited by their task representations, the dimensionality of the search space, and their ability to execute high-dimensional, complex tasks robustly. BELT approaches such problems through minimal constraints and structure on the task space, instead relying on learned local policies to guide the search towards task relevant regions.

*Model-Free Learning* offers a promising direction to execute such tasks, though theses methods have difficulty reasoning over long horizons. To enable long-horizon performance, previous works often enforce hierarchy or embed policies within classical structure. Hierarchical reinforcement learning [4, 5, 18] works to temporally extend planning at lower levels, reducing the horizon the high-level policy must reason over [19]. Recent work on goal-conditioned reinforcement learning learns to efficiently achieve arbitrary goal states [20, 3]. This casts the problem for the high-level policy as finding a sequence of goal states [8, 9, 21, 22]. This also allows the embedding of policies in classical motion planning structures [23, 24, 25, 26, 27, 28, 29, 30]. Search on the replay buffer (SoRB) [27] for example builds a searchable graph of the replay buffer given a learned distance metric for a goal-conditioned policy. In contrast to SoRB and other previous work, BELT uses a tree search equipped with a task-conditioned model to enable searching for plans outside of previous experience and is capable of solving non-goal-conditioned tasks with its model-based success check.

*Model-based Planning* allows longer-horizon, sequential planning by moving reasoning to the model level. In this way planning can be performed by rolling out models and evaluating such rollouts, [31, 32]. To extend the performance of models to longer-horizons, models can be trained for multi-step prediction [33] and planned over [34]. Others use models within classical algorithms, such as sampling-based motion planning [35] or control [36, 37]. In this work, we leverage a task-conditioned model to temporally extend the model prediction over edges with consistent tasks, resulting in more robust long-range performance. We additionally use the model to broadly search the state space via an RRT-inspired search.

## 1.2  Statement of Contributions.

The contribution of this paper is an algorithm Broadly-Exploring, Local-policy Trees (BELT) that is capable of planning long-horizon, sequential tasks in complex environments. BELT leverages (**1** – Sections 2 and 2.2) the state space exploration abilities of planning through an RRT-inspired search, (**2** – Section 2.1) the local guidance of a learned task-conditioned policy, capable of executing complex tasks in high-dimensional spaces, and (**3** – Section 2.3) the long-horizon propagation and reasoning of a learned, temporally extended task-conditioned model. BELT is demonstrated with a learned goal-conditioned policy in a realistic environment to plan effectively, significantly outperforming baselines. Furthermore, these plans are directly executable by the policy.

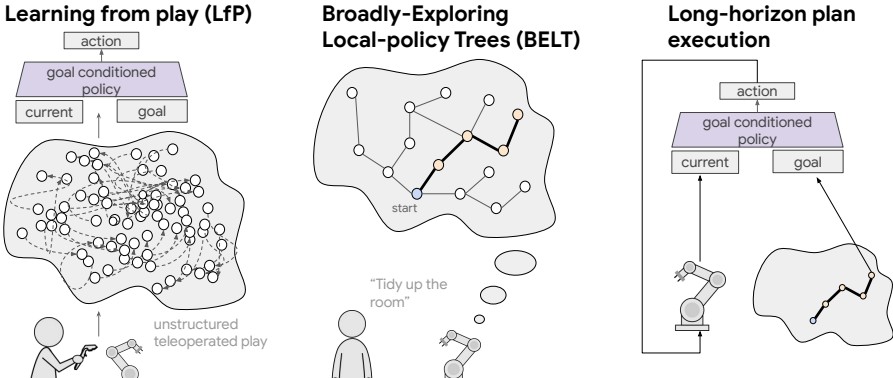

**Figure 2:** BELT with Play Latent Motor Plans [38] learns a general, goal-conditioned policy as well as a model from teleoperation data. Given a long-horizon task, BELT translates this into an RRT-inspired, model-based tree search through the space, where each edge represents a single task sampled from demonstration data. Trajectories are verified successful via a trajectory-wise success check on the model. Finally, the full long-horizon plan is executed online by the policy.

## 2 Broadly-Exploring, Local-policy Trees

We begin with a general outline of BELT and then discuss concrete implementations of algorithmic subroutines in Sections 2.1 - 2.3. Briefly, BELT searches for long-horizon, sequential plans via a model-based, task-conditioned tree search. This search is built on a backbone of an RRT with a local, learned task-conditioned policy. This leverages the efficient global-exploration of RRTs, the local directed actions of a learned task-conditioned policy, and the sequential reasoning of a model.

*Problem Statement.* In this work we refer to a task and task space in a general sense and we consider planning in environments which are predictable and known, with a fully observed state space $\mathcal{X}$. The only requirements of the task space $\mathcal{Z}$ is that it is sampleable and well-covers the space of useful actions. For example, this can be explicitly given as a set of tasks, implicitly learned via a latent task space, or structured as the state space $\mathcal{X}$ with a goal-conditioned policy. To allow for general task definitions, a success criteria Success is used that is considered a "black box". Given a trajectory $\{x_{\text{init}}, ..., x_T\}$, Success outputs if the trajectory satisfies all necessary conditions. This formulation allows for a general problem definition that includes goal-conditioned planning, but also allows for transient tasks that cannot be expressed by a goal state. The cost used in this work, referred to interchangeably with distance, is the time of a trajectory, though the approach is general to other costs. The environment considered in this work is continuous, known, and static, with the exception of motion as a result of interaction with the robot. The complexity of the environment arises generally from high dimensional states (e.g., many objects) and challenging interactions (e.g., grasping) and the sequentiality arises from tasks which are bottlenecked by previous task completion.

*BELT.* The search is outlined in Algorithm 1 and BELT is shown with progressively more concreteness in Figs. 1 - 3. Fig. 1 shows a hypothetical form of BELT with an explicit task space, Fig. 2 shows BELT with a goal-conditioned policy that is learned from play [38], and Fig. 3 shows BELT as applied to the experimental environment used herein. A new problem is defined by an initial state, $x_{\text{init}}$, and success criteria Success. The search initializes a tree $\mathcal{T}$ with $x_{\text{init}}$ as its root (Line 1) and iteratively adds task-conditioned edges. For each iteration $i$, a state $x_{\text{expand}} \in \mathcal{T}$ within the tree is selected, along with a task $z_i \in \mathcal{Z}$ (Lines 3-4). BELT requires a sampleable task space $\mathcal{Z}$ and task-conditioned policy, $\pi(a_t|x_t, z_i)$, which outputs an action $a_t$ at time $t$ given the current state $x_t$ and task $z_i$. The goal of the task space is to well cover the state space and guide the search towards useful actions; the task space and policy used herein are discussed in detail in Section 2.1. The goal of the selection of the tree node $x_{\text{expand}}$ is to guide the search towards unexplored regions and to exploit promising paths; this is discussed in detail in Section 2.2. Given the task $z_i$ and state $x_{\text{expand}}$, the policy is then propagated for $T$ timesteps via a model $p(x_{t+1}|x_t, a_t)$ to create trajectory $\{x_{\text{expand}}, x_1, ..., x_{T-1}, x_T\}$, saving the final state $x_{\text{new}} = x_T$ (Lines 6-7). The goal of the model is to predict trajectories given a task (in the absence of known dynamics); the model used herein is discussed in more detail in Section 2.3. In this way, each edge may be thought of having a temporally extended consistent intent, such as attempting a single task or reaching a goal. This edge is then added to the tree and the trajectory from $x_{\text{init}}$ to $x_{\text{expand}}$ to $x_{\text{new}}$ is checked for success with Success (Line 8). This search continues for $N$ iterations, at which time the lowest-cost path marked as successful is returned if it exists (Line 9).

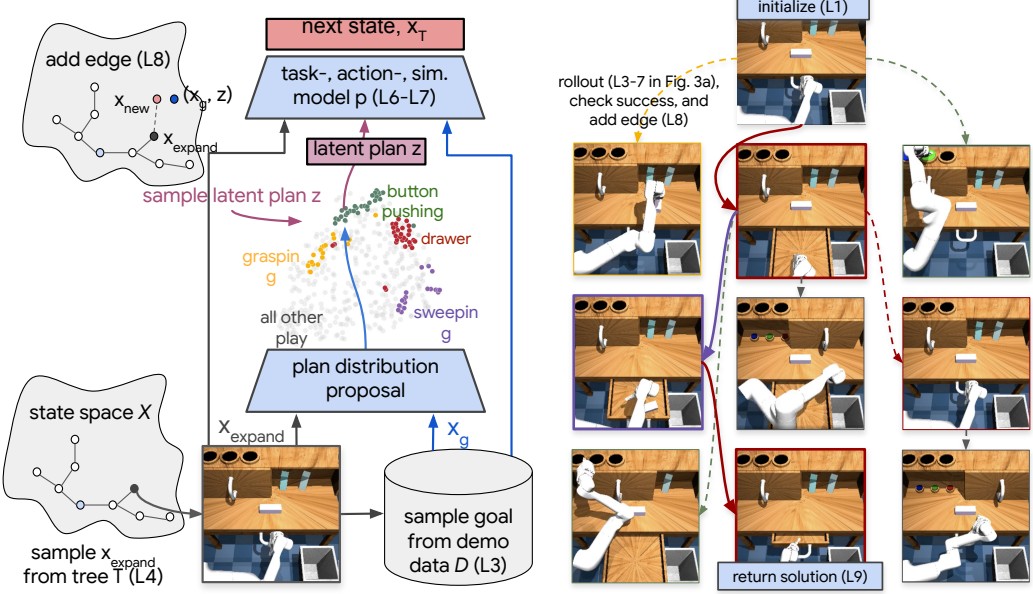

(a) An individual rollout of Alg. 1 with Play-LMP. Sample a goal state and latent "plan" (Line 3) and a tree state to expand from (Line 4). Policy takes as input a goal state, latent "plan", and the expansion state. This is then input to the model and rolled out (Lines 6-7.) The new trajectory is checked for success and added to the tree (Lines 8).

(b) Following Alg. 1, iteratively samples tree states and tasks, and rolls them out to add edges to a tree (Line 3-8). The solution trajectory is selected as the lowest cost solution, as verified by the binary success check Success (Line 9). After this offline planning, the sequence of subtasks is then executed online by the policy. Larger figure in Appendix Section 5.9.

Figure 3: Broadly-Exploring, Local-policy Trees with a Play-LMP policy (Section 2.1).

---

**Algorithm 1** Broadly-Exploring, Local-policy Trees

**Input:** initial state $x_{\text{init}}$, sample count $N$, and success check Success
1: Initialize tree $\mathcal{T}$ with state $x_{\text{init}}$.
2: **for** $i$ in $1 : N$ **do**
3:     Sample a task $z_i$ from task space $\mathcal{Z}$.                           ▷ Section 2.1
4:     Select a state $x_{\text{expand}}$ from $\mathcal{T}$ to expand from.          ▷ Section 2.2
5:     **for** $t$ in $1 : T$ **do**
6:         Run policy $a_t \sim \pi(a_t|x_t, z_i)$.                                  ▷ Section 2.1
7:         Rollout model $x_{t+1} \sim p(x_{t+1}|x_t, a_t)$.                         ▷ Section 2.3
8:     Update $\mathcal{T}$ with state $x_{\text{new}} = x_T$, trajectory $\{x_{\text{expand}}, ..., x_{\text{new}}\}$, task $z_i$, and Success($\{x_{\text{init}}, ..., x_{\text{new}}\}$).
9: **return** minimum time successful trajectory or failure if none

---

## 2.1 Learning a Task Space and Task-Conditioned Policy From Play

In this work we implement BELT with a general-purpose goal-reaching policy at the low level, called Learning from Play (LfP) [38]. To learn from "play" we assume access to an unsegmented teleoperated play dataset $\mathcal{D}$ of semantically meaningful behaviors provided by users, without a set of predefined tasks in mind. For example, a user might open and close doors, rearrange objects, or simply tidy the scene.

To learn control, this long temporal state-action log $\mathcal{D} = \{(x_t, a_t)\}_{t=0}^{\infty}$ is relabeled [3, 20], treating each visited state in the dataset as a "reached goal state", with the preceding states and actions treated as optimal behavior for reaching that goal. Relabeling yields a dataset of $D_{\text{play}} = \{(\tau, x_g)_i\}_{i=0}^{D_{\text{play}}}$, where each example consists of a goal state $x_g$ and a demonstration $\tau = \{(x_0, a_0), ...\}$ solving for the goal. These can be fed to a simple maximum likelihood goal conditioned imitation objective, $\mathcal{L}_{\text{LfP}} = \mathbb{E}_{(\tau,\ x_g) \sim D_{\text{play}}} \left[ \sum_{t=0}^{|\tau|} \log \pi_\theta (a_t|x_t, x_g) \right]$, to learn policy $\pi_\theta (a_t|x_t, x_g)$, parameterized by $\theta$. The motivation behind this collection is to allow play to fully cover the state space using prior human knowledge, then to distill short-horizon behaviors into a single goal-directed policy.

We could in principle use any goal-directed imitation network architecture to implement the policy $\pi_\theta (a_t|x_t, x_g)$. For direct comparison to prior work, we use Play Supervised Latent Motor Plans (Play-LMP) [38]. Play-LMP addresses the inherent multimodality in free-form imitation datasets. Concretely it is a sequence-to-sequence conditional variational autoencoder (seq2seq CVAE), au-

toencoding contextual demonstrations through a latent "plan" space. The decoder is a policy trained to reconstruct input actions, conditioned on state $x_t$, goal $x_g$, and an inferred plan $z$ for how to get from $x_t$ to $x_g$. At test time, Play-LMP takes a goal as input, and infers and follows plan $z$ internally. A t-SNE of this plan space is shown in Fig. 3a. See [38] for details. With Play-LMP as our policy, the task space $\mathcal{Z}$ is the union of the goal $x_g$ and the latent plan $z$.

## 2.2 Biasing the Search

Given a goal-conditioned policy, as Play-LMP is, RRT [1] is used with a few adaptions to broadly search the state space, chose achievable edge tasks, and bias towards low-cost paths (Fig. 4a).

To broadly search the state space, a key component of RRT's ability to efficiently explore is what is known as the Voronoi bias. For an RRT, this bias states that at each iteration, the selection of the tree node for expansion is proportional to the volume of its Voronoi region, meaning nodes in larger unexplored regions are more likely to be selected and thus RRTs rapidly explore [1]. To select a node in RRT, a sample is drawn from the state space and the nearest node within the tree is selected as the expansion node. For BELT, we seek to leverage a Voronoi bias via a temporal distance $\Delta t$ between states $x_i, x_j \in \mathcal{X}$ as the L2 distance is not a good distance for this problem. For example, consider the block in Fig. 3b. If it is inside the drawer or on top of the desk, the L2 distance is quite small, but to connect these two states requires opening the drawer, grasping the block, placing the block, and closing the drawer – a significant distance. As the temporal distance is not easily quantifiable, we learn to predict it with a temporal distance classifier, $\tau_{\text{tdc}}(x_i, x_j)$, with $K$ exponential categories corresponding to temporal distance intervals, $d_k \in \{[0,1), [1,2), [2,4), [4,8), [8,16), ..., [128, 256)\}$ based on [39]. The temporal distance classifier is implemented as a neural network and trained over policy rollouts to predict the distribution over class labels via a cross-entropy classification loss, $\mathcal{L}_{\text{tdc}} = -\sum_{l=1}^{K} y_l \log(\hat{y}_l)$ with $\hat{y} = \tau_{\text{tdc}}(x_i, x_j)$, where $y$ and $\hat{y}$ are the true and predicted label distributions respectively. Briefly, we note that we also considered a temporal distance regressor, but found it considerably less accurate.

The selection of an expansion state thus proceeds by first selecting a goal state from a demonstration dataset from rollouts of the policy $x_{\text{sample}} \in \mathcal{D}_\pi$. The temporal distance is then predicted between $x_{\text{sample}}$ and each node in the tree. All nodes below a cutoff temporal distance $d_{\text{cutoff}}$ are then added to a set of possible nodes to expand from. Within this set the node with the lowest-cost to come is selected for expansion, $x_{\text{expand}}$. This biases the tree search towards nodes that are capable of reaching the goal state and towards nodes with efficient trajectories. It further allows a Voronoi bias by temporal distance. The value of $d_{\text{cutoff}}$ represents a trade-off between exploiting more promising paths (larger values of $d_{\text{cutoff}}$) and exploring new ones (smaller values of $d_{\text{cutoff}}$); this trade-off is studied experimentally in the Appendix Section 5.3. This procedure modifies lines 3 and 4 of Algorithm 1.

We note that a common and effective approach to biasing search in sampling-based motion planning is to goal bias, i.e., biasing samples towards states within the goal region. As success often requires full sequences of trajectories rather than individual states we do not generally have access to goal states in this problem setting and thus we do not use goal biasing of this form.

## 2.3 Learning a Task-Conditioned Model

Given a policy, the tree search plans via a model. In this work we consider three options for models: a simulator, an action-conditioned model, and a task-conditioned model, each shown in Fig. 4b and 4c. Access to a simulator or the exact dynamics is assumed for much of the motion planning community when planning (unless explicitly planning under uncertainty). This acts as an upper bound herein and tests the ability of BELT to explore the state space and plan sequential tasks given a perfect model. It also shows bounds on the robustness of our policy. The action-conditioned model learns to predict the next state $p_{\text{action}}(x_{t+1}|x_t, a_t)$ given the current state $x_t$ and action $a_t$. For a given trajectory, this is recursively applied for $T$ timesteps. The task-conditioned model learns a prediction of where the robot will be after $T$ timesteps, i.e., $p_{\text{task}}(x_{t+T}|x_t, z)$. This accounts for the closed-loop nature of the policy for a fixed task and temporally extends the prediction to reduce recursive errors. For the Play LMP policy, the task conditioning $z$ corresponds to a goal state and a latent plan. Each model is represented as a neural network and trained from a dataset of demonstration policy rollouts to minimize the L2 reconstruction loss. For a given rollout for task $z$, states $\{x_0, x_1, ..., x_T\}$, and actions $\{a_0, a_1, ...a_{T-1}\}$, the action-model seeks to minimize $\mathcal{L}_{p_{\text{action}}} = \sum_{t=0}^{T-1} ||\hat{x}_{t+1} - x_{t+1}||$ with $\hat{x}_{t+1} \sim p_{\text{action}}(x_t, a_t)$, while the task-model seeks to minimize $\mathcal{L}_{p_{\text{task}}} = ||\hat{x}_T - x_T||$ with $\hat{x}_T \sim p_{\text{task}}(x_0, z)$.

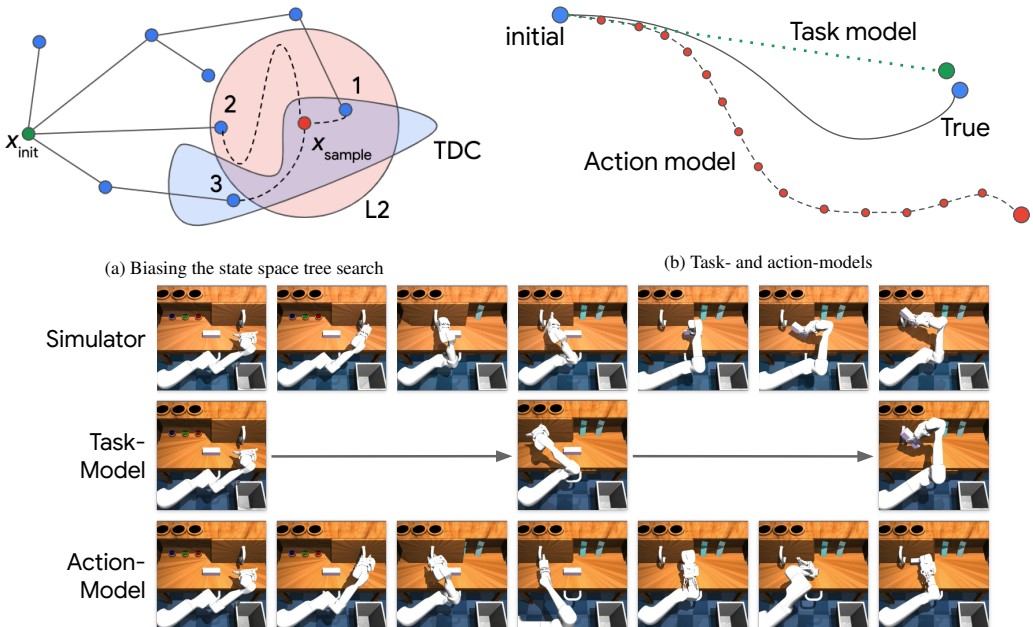

(a) Biasing the state space tree search

(b) Task- and action-models

Simulator

Task-Model

Action-Model

(c) Task-models, action-model, and simulator our experiment environment (see video link). Larger figure available in Appendix Section 5.9.

Figure 4: (4a) shows the bias used by BELT to choose the tree state $x_{expand}$ once $x_{sample}$ has been sampled. The blue region shows how the temporal distance between states may differ from the L2 distance, necessitating learning a temporal distance classifier. The choice between state 1 and 3 demonstrates the bias towards lower cost paths: though state 1 is closer to $x_{sample}$, node 3 has a much lower cost to come, and thus state 3 is selected. (4b-4c) shows the two types of models used in this work, an action- and a task-conditioned model. The action-conditioned model often exhibits compounding errors as it is recursively applied along the trajectory, while the task-conditioned model avoids this by temporally extending the prediction and conditioning on the fixed task for the edge. This compounding error can be seen in the second task (block lifting) where the action-model becomes unstable and the end effector flails (see video link).

# 3   Experimental Results

The experiments herein are based on a simulated "playground environment" [38] (Discussed in more detail in the Appendix and shown in Appendix Fig 7) built in Mujoco [40]. The robot is a position-controlled 8-DOF robotic arm and gripper. The robot receives as observations at each timestep of the cartesian position and orientation of all objects including its end effector. The agent performs high-frequency, closed-loop control, sending continuous position, rotation, and gripper commands to its arm at 30 Hz. The environment is known and static (unless the direct result of interaction with the robot) and contains a desk with a sliding door and drawer that can be opened and closed as well as a movable rectangular block and 3 buttons that can be pushed to turn on different colored lights (for a total of 19 continuous dimensions, detailed in full in Appendix 5.1).

*Long-horizon evaluation.* We evaluate our model and its baselines on large set of challenging long-horizon manipulation tasks as in [41], which require several subtasks to be executed in order. Here, multi-stage tasks are constructed by considering all valid N-stage transitions between a set of 18 core short-horizon manipulation tasks. See [38, 41] for a complete discussion of the 18 tasks, which span different families, e.g. pushing buttons, opening and closing doors/drawers, grasping and lifting a block, etc. We evaluate on "Chain-4", which results in 925 unique task chains (Appendix Fig. 7).

Note that prior work [41] assumes a human is in the loop providing each next subtask to the agent in the sequence. In our setup, an agent must provide *itself* subgoals to achieve the multi-stage behavior. Other prior long-horizon evaluations, such as [9], describe the long-horizon task as a final goal state to reach. While this valid for many useful manipulation tasks, many are impossible to infer from the final goal state alone. For example, in this environment, when a button is no longer pressed down, the corresponding light turns off. In this way, the final state for task "1) open drawer, 2) push green button, 3) close drawer" may appear to an agent as "no change to the environment". We consider a more general task specification scheme—an indicator function, defined over the full agent trajectory, which returns true if the full desired sequence was executed correctly and false otherwise. We refer to this here as the "success detector", which our high-level agent plans against.

## 3.1 Algorithmic Parameters and Baselines

*Baselines.* We compare against two baselines that use the same underlying Play-supervised Latent Motor Plan (Play-LMP) [38] policy. We compare to a model-based planner (cross entropy method (CEM) [42]) and a Play-LMP with given goal state. CEM was implemented in the task space to select four sequential tasks. The tasks were then rolled out with the simulator and task-model. The distribution was then refit to successful trajectories. We also compare to a Play-LMP policy that is given a single goal-state that satisfies all the subtasks (this single goal state existed for 36% of the chains). This baseline demonstrates that a simple approach of using long-horizon goals with a policy trained on short-horizons performs poorly and that BELT is able to plan long-horizons with local policies. Finally, we also compare to an "oracle" LMP policy, Chain-LMP, which is given the four chain tasks in a row, demonstrating an upper bound of what the BELT plan may achieve given the performance of the underlying policy.

*Parameters.* The low-level policy is trained with the same play logs collected in [38]; $\sim$7h of teleoperated play relabeled into $D_{\text{play}}$, containing $\sim$10M short-horizon demonstrations, each 1-2 seconds (see video for examples of this data). The TDC and models were trained on demonstration data for $\sim$ 100k rollouts. $\tau_{\text{tdc}}$ was trained on this dataset to an accuracy of 82% and $d_{\text{cutoff}}$ was set to $< 64$ based on analysis in the next section. The edges were rolled out with a randomly drawn timestep of $T \in \{32, 64, 96\}$, as inspired by [10]. BELT used 2500 goal samples drawn from demonstration data of policy rollouts, rather than the play data generated by human operators. We note that we also tried samples from the play dataset, but found many of the goals unreachable by the policy, and overall worse performance. See the Appendix for detail on parameters and tradeoffs.

*Metrics.* We evaluated the performance of BELT across three metrics intended to capture both its ability to plan and how robust those plans are. "Solution Found" is the rate at which the planner returns that it has found a solution that satisfies the success criteria, capturing an algorithm's ability to explore and plan. "Success Rate" is the rate at which the solution is successfully executed if it is found (in this work we rollout successful solutions nine times), capturing how robust the plan is. This robustness may be with respect to a stochastic policy, errors in the learned model, or a stochastic environment. "Feasibility" is the rate at which any of the replays for a given solution was able to solve the problem, capturing whether the plan was feasible (partially isolating the impact of an inconsistent policy). These are discussed in more detail in Appendix Section 5.7.

## 3.2 Results

We begin by comparing the overall performance of BELT, different models, and the baselines in Table 1 on the chain tasks and Fig. 5 shows several planned trajectories. We study each algorithm's performance on the axis of finding solutions (based on the algorithm output) and on robustness. Chain-LMP shows the upper bound of robustness of the policy to repeat four successive tasks. CEM with the simulator and LMP with a single-goal state fail to solve most chains. CEM with the model solves more tasks, but the solutions are not robust as the search seems to exploit inaccuracies in the model. BELT with the action model too exploits model inaccuracies – the model quickly becomes unstable as it is recursively rolled out over every timestep. Solutions are found for all chains, but the solutions are not robust or feasible for most, possibly due to impossible states input to the success detectors. The task model performs much better due to its temporally extended prediction, though we note multistep predictions hold promise to improve the action-conditioned results [33].

BELT with the task model and the simulator perform well, solving 66% and 82% of the problems respectively, and replay at similar rates and only slightly worse than the oracle. This demonstrates BELT's ability to both search the space of solutions and find feasible plans. The task model solves 16% fewer problems than directly using the simulator.

| Algorithm | Model | Solution Found | Success Rate | Feasible |
|---|---|---|---|---|
| Chain-LMP (Oracle) | – | – | 32% | 87% |
| LMP | – | – | 6% | 21% |
| CEM | Simulator | 3% | 40% | 58% |
| CEM | Task-Model | 11% | 7% | 15% |
| BELT | Action-Model | 100% | 3% | 8% |
| BELT | Task-Model | 66% | 27% | 56% |
| BELT | Simulator | 82% | 26% | 68% |

Table 1: Success rate and robustness for BELT, CEM, and LMP. Plan robustness is measured by the percent of solutions executed successfully (success rate) and if there was at least one successful in nine attempts (feasible). Note that LMP is only evaluated on chains that can be represented by a single goal state (36% of the chains). All approaches evaluated on only this set (which are comparatively easier) is shown in the Appendix Table 3.

| Chain Contains Task | Solution Found | | | Feasible | | |
|---|---|---|---|---|---|---|
| | BELT Simulator | BELT Task-Model | Ratio | BELT Simulator | BELT Task-Model | Ratio |
| overall | 82% | 66% | 0.81 | 68% | 56% | 0.82 |
| button, push any | 73% | 54% | 0.74 | **63%** | **32%** | **0.50** |
| no buttons | 99% | 89% | 0.90 | **76%** | **86%** | **1.12** |
| drawer open/close | 82% | 70% | 0.85 | 69% | 64% | 0.93 |
| grasp | 92% | 77% | 0.84 | 73% | 70% | 0.95 |
| knock | 91% | 71% | 0.78 | 58% | 55% | 0.95 |
| rotate left | **93%** | **0%** | **0.00** | **38%** | – | – |
| rotate right | 89% | 78% | 0.88 | 77% | 74% | 0.97 |
| shelf, in/out | 77% | 56% | 0.72 | 57% | 54% | 0.95 |
| slider, open/close | 81% | 63% | 0.78 | 67% | 58% | 0.87 |
| sweeping | 76% | 64% | 0.84 | 65% | 76% | 1.17 |

Table 2: Task performance breakdown. The task-model particularly has difficulty planning "rotate left" tasks due to an angle discontinuity, and the policy has trouble executing the task for the same reason. With button presses both approaches plan poorly (BELT with a simulator solves 99% without and 73% with) and the task-model's plans execute poorly, because the button tasks are transient and thus often do not appear in goal states.

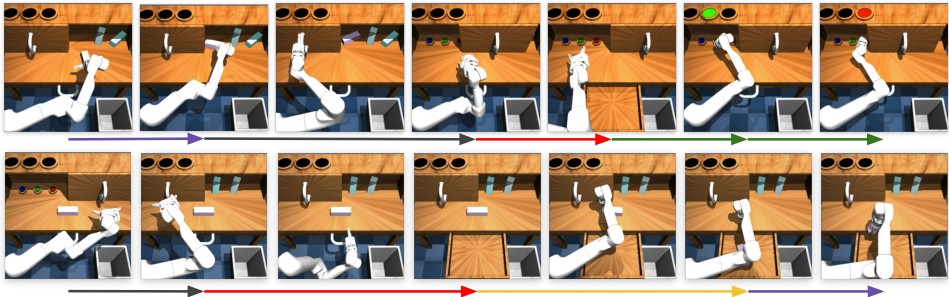

Figure 5: Plans from BELT, demonstrating its ability to plan long-horizon, sequential trajectories (video link).

In Table 2, we analyze which tasks proved most difficult for BELT, and particularly where the task-conditioned model and the simulator performance most diverged. For both the simulator and task-model, the button presses proved to be part of the most difficult chains to plan and execute. This is a because the button presses are transient and do not leave a permanent state change, which is particularly challenging for a goal-conditioned low-level policy to well cover. The task-model has difficulty with chains with the "rotate left" task on the block, while the simulator plans quite well over them but fail to execute many. This is a result of the discontinuity of angles, i.e., if the block starts at 0 radians and is rotated left, the angle jumps to $2\pi$ radians, which is particularly difficult for the model. This indicates a need for an angular representation such as quaternions or axis-angles.

## 4 Conclusion and Future Work

*Conclusions.* This work presented BELT, an algorithm capable of planning long-horizon tasks. BELT leverages the rapid state space exploration of RRT, the robust and general local performance of learned task-conditioned policies, and the long-horizon reasoning of a task-conditioned model. It is demonstrated in a realistic, complex, manipulation environment with a goal-conditioned policy to be capable of planning consistently and robustly.

*Real-world Experiments.* This method builds upon a scalable approach to learning goal-conditioned low-level policies ([38]), which were demonstrated to yield high success rate on the 18-task benchmark using onboard sensors only. While these results were obtained in simulation, the scalability of the local policy learning makes real-world deployment promising. Our work also relies on task success detectors, real-world experiments could make use of learned success classifiers such as [43]. Hence we estimate that there is a reasonable and scalable path to real-world experiments and plan to explore this in future work.

*Future Work.* In the future we wish to show how BELT can be extended well beyond the dataset and to other task representations, state representations (e.g., images), and models. We further wish to study how it can be used in replanning, particularly how the tree may be reused, as discussed in Appendix Section 5.8. We also wish to study how BELT can be used with constraints on obstacles, states, or model feasibility. Finally, we wish to study methods for generating not just any feasible plan, but robust and efficient plans.

## Acknowledgments

The authors wish to thank Karol Hausman, Vincent Vanhoucke, Alexander Toshev, and Aleksandra Faust for helpful discussions.

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
