# OpenReview forum: "Broadly-Exploring, Local-Policy Trees for Long-Horizon Task Planning"
_robot-learning.org/CoRL/2021/Conference — CoRL2021 Poster_

### Official Review · Reviewer_FQLy · 2021-07-23

**Originality:** Good
**Technical Quality:** Good
**Clarity Of Presentation:** Fair
**Impact:** 2

**Recommendation:**

Weak Accept: I recommend accepting the paper, but will not argue for my recommendation if the majority of other reviewers have a different opinion.

**Summary:**

This paper proposes a framework that combines imitation learning and planning approaches to solve long-horizon control tasks. Specifically, a goal-conditioned model is trained via imitation learning to achieve short-horizon behaviors, and an RRT planner uses this goal-conditioned model to find a sequence of temporally abstract waypoints to solve long-horizon tasks. Experiments on a simulated playground environment demonstrate that the proposed approach can solve long-horizon tasks more effectively than a pure goal-conditioned baseline and a variant using an alternative planning method.

**Issues:**

See the list in the “Strengths and Weaknesses” section.

**Reviewer Expertise:**

Very good: Comprehensive knowledge of the area

**Strengths And Weaknesses:**

The overall goal of this paper — solving long-horizon tasks — is motivated well in the introduction, and the related work section sets the context for relevant work well. The first three figures — showing a progressively more concrete overview of the method — also complement the paper well. These components all help to understand the high-level ideas of the paper. That said, there are a number of areas for improvement regarding the clarity and technical quality of the methods and experiments section:
1. The definition of the goal-conditioned model is a bit unclear. The paper mentions that the underlying goal-conditioned policy is based on Play-LMP, which consists of both a goal state $g$ and a task vector $z$, yet in this paper the policy is only conditioned on $z$.
2. The metrics in the main experimental results (table 1) should be defined more clearly. It is possible to infer the distinction between “Solution Found” and “Success Rate” but these details should not be up to interpretation.
3. How long does it take the planner to find a solution? This is an important consideration if we seek reactive agents that can operate in real-time.
4. What is the dimensionality of the state space? This can go in the appendix.
5. Is the underlying Play-LMP policy trained on short-horizon windows of (state, goal) pairs or arbitrary horizons? If it is trained for short-horizon windows only then it makes sense why the LMP baseline performs poorly — in that case the LMP comparison would not be fair because the baseline was not trained to solve long-horizon tasks.
6. The LMP baseline is only evaluated on tasks that can be expressed as a single goal state, in contrast with the other baselines. It may be that single-goal state tasks entail dealing with a unique set of challenges, making this an unfair comparison.
7. Clarify the distinction between the play dataset and the demonstration dataset. Why is a separate demonstration dataset needed?
8. In the cases when the method does not find a solution, what are the underlying reasons behind these cases? Expand on the analysis.
9. It is entirely possible that the dynamics model predicts infeasible states, eg. when two blocks penetrate with each other. The success function may erroneously predict success for infeasible states, unless it can filter such states to start with. How does the success function deal with such states?
10. The paper claims that the proposed method can deal with high-dimensional states. Have you considered tasks operating over images? Are there specific modifications you need to make to your framework to deal with images?

**Summary Of Recommendation:**

This paper is well-motivated and the high-level idea of the paper is easy to understand. There are however a number of low-level details that are unclear in the current iteration of the paper. Refer to the “Issues” section for details. These issues are hopefully easy to address, **at which point I will raise my recommendation.**

---
Update: raised recommendation to weak accept, **but with some hesitation.** Please refer to my followup post for details.

---

> ### Author Response · Authors · 2021-08-29
> **Metrics, state spaces, and new ablations over tasks**
>
> Thank you for your valuable feedback. We appreciate that you found the paper well motivated and clear. We respond to your comments individually below, and have made revisions to the paper (highlighted in blue) based on your comments, which we believe has improved the paper. Please let us know if you have any remaining concerns or questions!
>
> > 1. The definition of the goal-conditioned model is a bit unclear. The paper mentions that the underlying goal-conditioned policy is based on Play-LMP, which consists of both a goal state g and a task vector z, yet in this paper the policy is only conditioned on z.
>
> BELT is capable of planning with a very general notion of task space, including an explicit set of tasks, a goal-conditioned policy, latent spaces, and more. We use Play-LMP as our underlying policy herein, and with Play-LMP this task space is defined as both a goal state and a latent plan. We have added the following to clarify this: “With Play-LMP as policy, the task space Z is the union of the goal x_g and the latent plan z.”
>
>
> > 2. The metrics in the main experimental results (table 1) should be defined more clearly. It is possible to infer the distinction between “Solution Found” and “Success Rate” but these details should not be up to interpretation.
>
> We have significantly expanded our metrics discussion in Section 3.1 and in Appendix Section 5.7. We have added formal definitions and long form discussion in Appendix Section 5.7. Results Section 3.1 now states: ““We evaluated the performance of BELT across three metrics intended to capture its ability to plan and how robust those plans are.  "Solution Found" is the rate at which the planner returns that it has found a solution that satisfies the success criteria, capturing an algorithm's ability to explore and plan. "Success Rate" is the rate at which the solution is successfully executed if it is found (in this work we rollout successful solutions nine times), capturing how robust the plan is. This robustness may be with respect to a stochastic policy, errors in the learned model, or a stochastic environment.  "Feasibility" is the rate at which any of the replays for a given solution was able to solve the problem, capturing whether the plan was feasible (partially isolating the impact of an inconsistent policy). These are discussed in more detail in Appendix Section 5.7.”
>
> > 3. How long does it take the planner to find a solution? This is an important consideration if we seek reactive agents that can operate in real-time.
>
> At this time BELT is not near real-time, though some optimizations, such as parallelizing rollouts, tree reuse, and more streamlined code may allow it to approach replanning. BELT with the simulator solved problems in ~10 minutes, while BELT with the task model took ~2 minutes. We have moved some discussion of replanning through reuse of the tree into Appendix Section 5.8.
>
> > 4. What is the dimensionality of the state space? This can go in the appendix.
>
> The total dimensionality is 19. We have added “(for a total of 19 continuous dimensions)” to Section 3 and in the Appendix:
> “The full state space is 19-dimensional and contains:
> - a robot arm and gripper (8 dimensional position, orientation, and the amount each finger is closed)
> - a block (6 dimensional position and orientation)
> - a sliding door (1 dimensional continuous position)
> - a drawer door (1 dimensional continuous position)
> - three buttons (1 dimensional each, continuously tracking the amount pressed)“
>
> > 5. Is the underlying Play-LMP policy trained on short-horizon windows of (state, goal) pairs or arbitrary horizons? If it is trained for short-horizon windows only then it makes sense why the LMP baseline performs poorly — in that case the LMP comparison would not be fair because the baseline was not trained to solve long-horizon tasks.
>
> It is correct that Play-LMP was only trained on short-horizon windows and thus is not expected to perform well. We included Play-LMP primarily to show that the simplest possible approach of giving it a longer goal will not work and to show that BELT is able to extend the horizon of policies well. We have amended the text to reflect this: “This baseline demonstrates that a simple approach of using long-horizon goals with a policy trained on short-horizons performs poorly and that BELT is able to plan long-horizons with local policies.”

---

> > ### Author Response · Authors · 2021-08-29
> > **Part 2**
> >
> > > 6. The LMP baseline is only evaluated on tasks that can be expressed as a single goal state, in contrast with the other baselines. It may be that single-goal state tasks entail dealing with a unique set of challenges, making this an unfair comparison.
> >
> > Thank you for this observation. We have gone back and looked at only tasks in common and found that generally these problems are easier to solve and that the performance of BELT and CEM improves when only run on the subset of problems which could be expressed as a single state. The table below shows the results on just these problems and has been added to the Appendix, Table 3 (along with a reference note in Table 1).
> >
> > | Algorithm | Model | Solution Found | Success Rate | Feasible |
> > |---|---|---|---|---|
> > | LMP | - | - | 6% | 21% |
> > | CEM | Simulator | 7% | 38% | 56% |
> > | CEM | Task-Model | 11% | 18% | 35% |
> > | BELT | Action-Model | 100% | 5% | 14% |
> > | BELT | Task-Model | 84% | 43% | 83% |
> > | BELT | Simulator  | 98% | 33% | 75% |
> >
> > > 7. Clarify the distinction between the play dataset and the demonstration dataset. Why is a separate demonstration dataset needed?
> >
> > The demonstration dataset is created by rollouts of the policy, while the play dataset is human collected. For sampling during the search procedure, this keeps the search on the manifold of what the policy can accomplish. We have added discussion to the Parameters Section 3.1 to this effect. Furthermore, this demonstration dataset is necessary for the learned task-conditioned model. As our task-conditioned model is conditioned on goal g and latent z and the policy is learned via hindsight relabeling (such that x_T = g), learning a task-conditioned model on the play data would just learn identity, i.e., a network that directly passed through the input goal g.
> >
> > > 8. In the cases when the method does not find a solution, what are the underlying reasons behind these cases? Expand on the analysis.
> >
> > We have added a section to the appendix (Section 5.4) to discuss the convergence of BELT as well as Figure 9, which shows the success rate versus number of samples for BELT. Given the simulator, we found that at 2500 samples BELT is still solving more chains of tasks and may in the limit converge to a high solution rate. The problem and state space considered is quite complicated even with a good underlying policy, so this is not unexpected. Interestingly, of these still to be solved, almost all included button presses (99% of the problems without button pushes were solved). This is because the button press is a transient task, so sampling a goal state with it explicitly pressed is much less likely than other tasks. This could be remedied by rebalancing the dataset or biasing the search towards these states.
> >
> > We have added a row to Table 2 detailing this and discussion in the table caption:
> >
> > | Chain Contains Task | Solution Found | Solution Found | Solution Found Ratio | Feasible | Feasible | Feasible Ratio |
> > |---|---|---|---|---|---|---|
> > |  | BELT Sim | BELT Task Model |  | BELT Sim | BELT Task Model |  |
> > | button, push any  | 73% | 54%	| 0.74 | 	63% | 32% | 	0.50 |
> > | no buttons  | 99% | 89%	| 0.90 | 	76% | 86% | 	1.12 |
> >
> > The task-model sees a similar curve, but shifted downward. This is primarily due to errors in modeling two tasks: “rotate left” and button presses. The “rotate left” was due to the discontinuity between 2pi and 0, and motivates particularly better angle representations. The aforementioned transient issue for button presses is doubly important for the task-conditioned model, which does not recreate the full trajectory, but rather just the end state. This could be remedied through either a model that recreates the full trajectory or a model that also predicts task completion.

---

> > > ### Author Response · Authors · 2021-08-29
> > > **Part 3**
> > >
> > > > 9. It is entirely possible that the dynamics model predicts infeasible states, eg. when two blocks penetrate with each other. The success function may erroneously predict success for infeasible states, unless it can filter such states to start with. How does the success function deal with such states?
> > >
> > > This indeed can cause erroneous successes while planning and may be a factor in the reduced feasibility of BELT + Task-Model (and certainly in BELT + Action-Model). Though in practice we find a relatively modest reduction in feasibility with the task-model, the main culprit is tasks with button pushes. This task is challenging due to its transient nature, but also as it is difficult to model the gripper-button interaction (which we have added a comment to that effect). We believe these ideas particularly motivate further research into learned models with constraints on feasibility or planning with feasibility. This may particularly be a good fit for RRT-style approaches as they are often used for planning in the presence of constraints on collisions with obstacles. We have added some discussion to future work to this effect and in the result section.
> > >
> > > > 10. The paper claims that the proposed method can deal with high-dimensional states. Have you considered tasks operating over images? Are there specific modifications you need to make to your framework to deal with images?
> > >
> > > Indeed, we believe that images are a very interesting next step for this work and within the realm of what BELT can deal with. However, two key additional challenges exist for such an approach: success detection on images and a good enough model. For success detection, the current version of BELT uses explicit success functions, but that becomes more difficult on images and particularly on images generated via a model. Likely one would need to learn success detectors and also train them on model generated images. For the image, a high capacity network may be able to generate reasonable images, but generating images recursively through the model adds additional challenges. The task-conditioned, temporally-extended model alleviates some of this challenge, but one may need techniques from the vision community like GANs to ensure high enough quality images. Another option is to construct a latent space that is jointly trained to reconstruct images and predict success and fully plan in that space. We have added images as future work to the conclusions.

---

> > > > ### Comment · Reviewer_FQLy · 2021-09-03
> > > > **Raising my recommendation, but with hesitation**
> > > >
> > > > I would like to thank the authors for their thorough rebuttal response. Many of my initial confusions have been addressed, and as I promised I am going to raise my recommendation to weak accept. However, **I did so with hesitation, as I think there are still some important concerns:**
> > > >
> > > > *  The LMP baseline is only trained on short-horizon windows, why not train it for long-horizon windows as well? It would make the comparison more fair and help understand whether planning is needed in the first place.
> > > > * Because the model has to hallucinate future outcomes, that leads to feasibility issues with unrealistic predictions, especially for images. It might be more scalable to use the existing states in the play/demo dataset as states to plan with, as done in Search on the replay buffer (SoRB).
> > > >
> > > > I encourage the authors to think about these issues and try to address them, even if the paper is accepted.

---

### Official Review · Reviewer_qgUz · 2021-07-24

**Originality:** Good
**Technical Quality:** Very Good
**Clarity Of Presentation:** Good
**Impact:** 4

**Recommendation:**

Strong Accept: I recommend accepting the paper and will argue for my recommendation even if other reviewers hold a different opinion.

**Summary:**

The authors propose a planning method for using imitation-learned policies to form plans over multiple sequential tasks. They use imitation learning to train a goal-based policy, and train a temporal distance classifier to predict the time to reach the sampled point. The proposed planning algorithm is similar to performing RRT in a space of task manipulations. They sample points from the dataset of demonstrations and attempt to reach those points, expanding the lowest-cost node on the tree that they predict can reach the point in under a certain time. They find that when using a simulator or task-conditioned model, they are frequently able to complete difficult strings of tasks that are difficult for existing methods to solve. They also find that the task-conditioned model is essential for preventing the planner from exploiting inaccuracies in the model and maintaining accuracy over long time horizons.

**Issues:**

It's not clear what the Chain-LMP method is or what it's doing. It's also not clear in what sense this method is an oracle, considering its low success rate. Another detail that I had difficulty making sense of was the difference between the success rate and the 'solution found' rate when using the simulator. It was my understanding that the experiments are performed on a simulator, so if the method finds a sequence of actions that solves the task with the simulator, why can't that sequence of actions just be directly applied to deliver a solution?

**Reviewer Expertise:**

Good: General knowledge of the area

**Strengths And Weaknesses:**

This method moves towards making it feasible to plan over sequences of tasks, which is a challenging problem. In particular, the complexity of the tasks that they attempt is impressive. I find it interesting as well that this is done without learning different networks for different tasks, instead learning by imitation without well-delineated 'tasks'.




**Summary Of Recommendation:**

The experimental results for this method are strong, and planning method combined with the goal-conditioned policies make sense as a general problem-solving tool for complicated complicated tasks. The actual planning algorithm bears very strong similarity to RL-RRT, so the interest here is primarily on the technical details needed to achieve these kinds of results. Luckily, the method delivers these technical details and results.

---

> ### Author Response · Authors · 2021-08-29
> **ChainLMP and metrics**
>
> Thank you for your positive feedback. We appreciate that you found the tasks “impressive”, the experimental results “strong”, and that the “method delivers these technical details and results”. We respond to your main comments individually below, and have made revisions to the paper (highlighted in blue) based on your comments, which we believe has improved the paper. Please let us know if you have any remaining concerns or questions!
>
> > It's not clear what the Chain-LMP method is or what it's doing. It's also not clear in what sense this method is an oracle, considering its low success rate.
>
> Chain-LMP is given a sequence of four subgoals that accomplish the chained tasks, thus it is the equivalent of an ideal / oracle planner. This allows us to isolate the loss in performance due to the underlying policy versus the planner (e.g., BELT). We have added the definition of Chain-LMP to the Baselines section, along with this intuition: “Finally, we also compare to an "oracle" LMP policy, Chain-LMP, which is given the four chain tasks in a row, demonstrating an upper bound of what the BELT plan may achieve given the performance of the underlying policy.”
>
> > Another detail that I had difficulty making sense of was the difference between the success rate and the 'solution found' rate when using the simulator.
>
> We have significantly expanded our metrics discussion in Section 3.1 and in Appendix Section 5.7. We have added formal definitions and long form discussion in Appendix Section 5.7. Results Section 3.1 now states: ““We evaluated the performance of BELT across three metrics intended to capture its ability to plan and how robust those plans are.  "Solution Found" is the rate at which the planner returns that it has found a solution that satisfies the success criteria, capturing an algorithm's ability to explore and plan. "Success Rate" is the rate at which the solution is successfully executed if it is found (in this work we rollout successful solutions nine times), capturing how robust the plan is. This robustness may be with respect to a stochastic policy, errors in the learned model, or a stochastic environment.  "Feasibility" is the rate at which any of the replays for a given solution was able to solve the problem, capturing whether the plan was feasible (partially isolating the impact of an inconsistent policy). These are discussed in more detail in Appendix Section 5.7.”
>
> > It was my understanding that the experiments are performed on a simulator, so if the method finds a sequence of actions that solves the task with the simulator, why can't that sequence of actions just be directly applied to deliver a solution?
>
> You are correct that one could directly replay the actions to replay the solution (to the limit of determinism of the simulator). However, the primary goal of BELT + simulator is to understand the exploration ability of BELT and how much is lost with action-conditioned and task-conditioned models. Each of these approaches (along with an accurate simulator) also holds promise in being applied in the real world and we expect generally that the closed loop rollout of the stochastic LMP will be more robust to any variations from a perfect replay.

---

### Official Review · Reviewer_yRzc · 2021-07-24

**Originality:** Good
**Technical Quality:** Fair
**Clarity Of Presentation:** Fair
**Impact:** 3

**Recommendation:**

Weak Reject: I recommend rejecting the paper, but will not argue for my recommendation if the majority of other reviewers have a different opinion.

**Summary:**

The paper presents a task and motion planning algorithm that combines the classical motion planning algorithm, RRT, and reinforcement learning to solve long-horizon problems. The authors have shown its superior performance over baselines in simulation.

**Issues:**

- As mentioned in Weaknesses, the motivation should be enhanced in comparison with literature on task and motion planners.
- I think the paper addresses planning only, not interleaved planning and execution. However, the right figure in Fig. 2 and images of intermediate steps in Fig. 3(b) might make readers think that the method proposes an interleaved planning and execution approach. Please clarify this.
- The motivation of employing RRT is not clear. RRT is used to find a trajectory that does not collide with obstacles in the high-dimensional configuration space. Are all x points in configuration space and does the Fig. 4(a) represent the configuration space? In its current form, it looks like RRT is used to find a trajectory in obstacle-free state space, which can also be done with simpler planners.
- Fig. 3(a) needs a huge improvement since it represent the main method. It would be helpful to indicate which part is connected to RRT and include used notations. Also, since there are the task space and the state space, please indicate which space it is in all figures.
- The state is not an image but the state of the robot and object as explained in the experiments. From figures, readers might think that the bird-eye view images are used as a state. It would be good to clarify what state is considered.
- Along with the above comment, the problem description is not clearly stated before explaining the method. Please clearly state what is known and unknown to the robot, and what are inputs (e.g. what kind of data) to the proposed method.
- In the below texts in page 3, it would be helpful to explain the overall method using the line number of Algorithm 1 and Figure 3(a). Currently, texts, figure and algorithm are disconnected.
- The authors used the narrow passage as a fundamental issue in classical planning but it is only a fundamental problem of sampling-based motion planners. There are other algorithms in classical planning, such as trajectory optimization, that does not have the narrow passage problem so this is incorrect.
- The theoretical results do not seem to add much value but interrupts the flow of the reading. I would suggest to take it out.
- The considered task in experiments involves with pushing buttons. Is the state that the button being pushed or not represented by a state x? This is a discrete variable so how can RRT handle discrete variables with other continuous variables in search?
- I understand there is a page limit so that including all experimental results is hard. Although the validation on entire task is critical, readers might also be interested in seeing how each component performs. For example, is x_sample obtained by RRT for a given x_goal semantically meaningful?; is sampled z from task space also semantically meaningful with respect to neighboring points in task space?

**Reviewer Expertise:**

Very good: Comprehensive knowledge of the area

**Strengths And Weaknesses:**

Strengths:
- The authors tackled an interesting problem that lies in the intersection of classical planning and end-to-end policy learning, which is gaining its popularity recently. I find the particular architecture of the proposed method novel.
- Many figures are included to help better understand the paper.
- The results seem promising.

Weaknesses:
- The motivation of the work should be improved in comparison with task and motion planning algorithms in literature. Task and motion planning algorithms can also solve the problems in known, quasi-static environments efficiently so it is not clear why one should use the proposed complex method that involves with RL and user inputs. More emphasis on complex tasks or giving a concrete example that cannot be solved without RL would be helpful.
- Some parts are not technically sound. Detailed comments are included in Issues below.
- The paper is not well readable. It includes heavy terminologies, which might be improved by unifying similar ones, and figures require more information, such as notations, explanation on different lines and colors used, and what space it represents. The placement of figures, algorithm, and texts needs to be revised so that texts match with the right figure and algorithm.

**Summary Of Recommendation:**

Although the paper proposes an interesting method in robot learning and planning, I find the paper hard to read and understand. Also, there are technically vague parts in the method that need to be clarified. Overall, the presentation needs an improvement. Especially, it is confusing how RRT interacts with RL setting, which is the most important part in the paper. Thus, I think the paper is on the borderline in its current form.

---

> ### Author Response · Authors · 2021-08-29
> **TaMP discussion, RRT motivation, Problem statement and state space, Remade fig 3**
>
> Thank you for your valuable feedback. We appreciate that you found the problem “interesting”, architecture “novel”, and results “promising”. We respond to your main comments individually below, and have made revisions to the paper (highlighted in blue) based on your comments, which we believe has improved the paper. Please let us know if you have any remaining concerns or questions!
>
> > - The motivation of the work should be improved in comparison with task and motion planning algorithms in literature. Task and motion planning algorithms can also solve the problems in known, quasi-static environments efficiently so it is not clear why one should use the proposed complex method that involves with RL and user inputs. More emphasis on complex tasks or giving a concrete example that cannot be solved without RL would be helpful.
> > - As mentioned in Weaknesses, the motivation should be enhanced in comparison with literature on task and motion planners.
>
> While we agree that many planning algorithms from task and motion planning are capable of solving these problems, they generally do so under a different set of assumptions. In particular, explicit tasks or symbols (such as pick <object> or place<object>) or explicit transition constraints are required in references 11-17. In contrast, BELT places very minimal constraints on the task space outside of success checks and allows the policy to guide the search towards task relevant regions. We have amended our comparison to task and motion planning to state: “Within robotics, as the robot must interact with the world, planning algorithms often consider the task and motion planning setting (contrasting with AI planning). Approaches in this setting generally plan in a hierarchical manner by defining explicit task-primitives, symbolic languages, or transition logic constraints. These methods then plan over them, for example, via trees or optimization-based methods [11, 12, 13, 14, 15, 16, 17]. These methods perform well in long-horizon planning, but are limited by their task representations, the dimensionality of the search space, and their ability to execute high-dimensional, complex tasks robustly. BELT approaches such problems through minimal constraints and structure on the task space, instead relying on a local policy to guide the search towards task relevant regions.”
>
> > - I think the paper addresses planning only, not interleaved planning and execution. However, the right figure in Fig. 2 and images of intermediate steps in Fig. 3(b) might make readers think that the method proposes an interleaved planning and execution approach. Please clarify this.
>
> We have clarified this with statements in the captions of each figure: “Finally, the full long-horizon plan is executed by the policy.” and “After this offline planning, the sequence of subtasks is then executed online by the policy. “
>
> > - The motivation of employing RRT is not clear. RRT is used to find a trajectory that does not collide with obstacles in the high-dimensional configuration space. Are all x points in configuration space and does the Fig. 4(a) represent the configuration space? In its current form, it looks like RRT is used to find a trajectory in obstacle-free state space, which can also be done with simpler planners.
>
> It is correct that all trajectories are assumed to be obstacle free as the goal states are from demonstration data and valid and the policy learns to avoid invalid states. However, due to its dimensionality, the object interaction, and the policy, complex dynamics and topologies are induced in the state space – complicating its search. We believe RRTs provide an efficient method for exploring such complicated state spaces, such as the original work which focuses heavily on searching with dynamical robots, e.g., [A](http://msl.cs.illinois.edu/~lavalle/papers/Lav98c.pdf) and [B](https://skat.ihmc.us/rid=1K7WQT337-XQJP8C-1YHM/Randomized%20Kinodynamic%20Planning.pdf). Furthermore, we believe that RRT is a relatively simple planner that requires few assumptions. However, we have added extensions to planning in the presence of constraints, whether they are constraints on the learned model or collisions to the future work: “We also wish to study how BELT can be used with constraints on obstacles, states, or model feasibility.”
>
> All points in x and 4a are in the state space (which in the examples used throughout is equivalent to the configuration space).

---

> > ### Author Response · Authors · 2021-08-29
> > **Part 2**
> >
> > > - Along with the above comment, the problem description is not clearly stated before explaining the method. Please clearly state what is known and unknown to the robot, and what are inputs (e.g. what kind of data) to the proposed method.
> >
> > We have added an explicit “Problem statement” section and the state space description “*Problem Statement*. In this work we refer to a task and task space in a general sense and we consider planning in environments which are predictable and known, with a fully observed state space X”. We have further clarified the state space below in response to this comment and the following.
> >
> > > - The state is not an image but the state of the robot and object as explained in the experiments. From figures, readers might think that the bird-eye view images are used as a state. It would be good to clarify what state is considered.
> > > - The considered task in experiments involves with pushing buttons. Is the state that the button being pushed or not represented by a state x? This is a discrete variable so how can RRT handle discrete variables with other continuous variables in search?
> >
> > We have amended the paper to clarify what the state representation is and that each of the state space dimensions is continuous. The buttons are represented by the amount they are pressed (a button push is considered successful if the button is pressed beyond a threshold). We have added “(for a total of 19 continuous dimensions)” to Section 3 and in the Appendix:
> > “The full state space is 19-dimensional and contains:
> > - a robot arm and gripper (8 dimensional position, orientation, and the amount each finger is closed)
> > - a block (6 dimensional position and orientation)
> > - a sliding door (1 dimensional continuous position)
> > - a drawer door (1 dimensional continuous position)
> > - three buttons (1 dimensional each, continuously tracking the amount pressed)“
> >
> > > - The paper is not well readable. It includes heavy terminologies, which might be improved by unifying similar ones, and figures require more information, such as notations, explanation on different lines and colors used, and what space it represents. The placement of figures, algorithm, and texts needs to be revised so that texts match with the right figure and algorithm.
> > > - In the below texts in page 3, it would be helpful to explain the overall method using the line number of Algorithm 1 and Figure 3(a). Currently, texts, figure and algorithm are disconnected.
> > > - Fig. 3(a) needs a huge improvement since it represent the main method. It would be helpful to indicate which part is connected to RRT and include used notations. Also, since there are the task space and the state space, please indicate which space it is in all figures.
> >
> > We have added Alg. 1 Line numbers to Figure 3, to the caption of Figure 3a and 3b, to the text description of BELT. We have also changed the notation in the figure to match the algorithm, indicating whether things are from the state space X or the latent task space Z, and added a tree figure matching the RRT expansion.
> >
> > > - The authors used the narrow passage as a fundamental issue in classical planning but it is only a fundamental problem of sampling-based motion planners. There are other algorithms in classical planning, such as trajectory optimization, that does not have the narrow passage problem so this is incorrect.
> >
> > We had been considering sampling-based motion planners as the state of the art for high-dimensional problems and thus considered narrow passages, but we agree these are not outside of the realm of approaches like trajectory optimization. Our key point was to state that these problems are often difficult due to their small and topologically complex goal regions, which may be narrow passages but can also be more general, and we have amended the paper to say this, stating: “presence of small, topologically complex goal regions, which naturally emerge in tasks that interact with the environment” in the abstract and “However, they have difficulty in the presence of complex, stochastic dynamics, high-dimensional systems, so-called narrow passages, and small, complex goal regions (which tasks naturally create).” in the introduction.
> >
> > > - The theoretical results do not seem to add much value but interrupts the flow of the reading. I would suggest to take it out.
> >
> > We have moved this section to the Appendix Section 5.6.

---

> > > ### Author Response · Authors · 2021-08-29
> > > **Part 3**
> > >
> > > > - I understand there is a page limit so that including all experimental results is hard. Although the validation on entire task is critical, readers might also be interested in seeing how each component performs. For example, is x_sample obtained by RRT for a given x_goal semantically meaningful?; is sampled z from task space also semantically meaningful with respect to neighboring points in task space?
> > >
> > > We build off the policy learned in [38], [Learning latent plans from play](http://proceedings.mlr.press/v100/lynch20a/lynch20a.pdf), for which FIgure 4 shows this analysis of z via a t-SNE latent space, which is found to semantically cluster, and which describes the policy and semantic meaning in significant detail. This figure is also included in Figure 3a of our work. In terms of sampling x_sample, these are sampled from a demonstration dataset of policy rollouts and may have semantic meaning. It is however possible that x_sample and x_expand have no tasks in between them or multiple tasks in between them.

---

> > > > ### Comment · Reviewer_yRzc · 2021-09-04
> > > > **Paper clarity has improved**
> > > >
> > > > Thank you for addressing my comments and for improving the paper. After reading the responses, I find the paper improved its clarity considerably, which was the major concern I had. Therefore, I am raising my recommendation to a **weak accept**.

---

### Official Review · Reviewer_Da4d · 2021-07-26

**Originality:** Good
**Technical Quality:** Very Good
**Clarity Of Presentation:** Good
**Impact:** 4

**Recommendation:**

Weak Accept: I recommend accepting the paper, but will not argue for my recommendation if the majority of other reviewers have a different opinion.

**Summary:**

The paper aims to solve long horizon state-based robotic tasks by leveraging RRT-style tree search with goal conditioned policy and task-dynamics model. The goal conditioned policy is learned from play-data using variational learning. It is able to solve subtasks with a shorter task horizon within a longer-horizon task. Simultaneously, the paper learns a task dynamics model which takes in latent task parameters to predict the final state after completion of the task. Finally, the paper uses a task dynamics model with RRT-style tree search to find the sequence of subtasks needed to complete the task and then solves each of the subtasks with the goal conditioned policy.

**Issues:**

Mentioned above. See the weakness section.

POST-REBUTTAL UPDATE: The authors have addressed my concerns. I am keeping my score to a weak accept. It's more of an accept for me (but the only other option available is strong accept)

**Reviewer Expertise:**

Good: General knowledge of the area

**Strengths And Weaknesses:**

Strengths:
The proposed technique is able to nicely solve hard long horizon tasks. For context, it is very hard to solve these tasks with existing deep RL (model free or model based) techniques with end to end training. The paper does a good job of leveraging past data to learn a goal conditioned policy as well as a task dynamics model and then using them to plan for longer horizon tasks.

Weaknesses:
The writing could have been a bit more clear with respect to the metrics reported. I don’t completely understand the difference between solution found and feasibility. By solution found, do we mean the agent is able to find some solution using the dynamics model (but that solution could be totally wrong if the dynamics model is bad)? If yes, when the simulator is used as the dynamics model, shouldn't the solution found be very similar to percentage feasibility? Also, if CEM finds a solution in 3% of the cases, how is the success rate 40%? It would be really nice to explain this clearly.


**Summary Of Recommendation:**

I like the use of goal conditioned policy, task dynamics model and RRT style tree search to solve hard long horizon tasks. That's why I am giving it a weak accept. I will be happy to increase the score if authors are able to more clearly explain the metrics used in reporting the results (as pointed above)

---

> ### Author Response · Authors · 2021-08-29
> **Significantly revised metrics description**
>
> Thank you for your positive feedback. We appreciate that you found that BELT “is able to nicely solve hard long horizon tasks” and well leverages past data. We respond to your main comments individually below, and have made revisions to the paper (highlighted in blue) based on your comments, which we believe has improved the paper. Please let us know if you have any remaining concerns or questions!
>
> > Weaknesses: The writing could have been a bit more clear with respect to the metrics reported. I don’t completely understand the difference between solution found and feasibility. By solution found, do we mean the agent is able to find some solution using the dynamics model (but that solution could be totally wrong if the dynamics model is bad)? If yes, when the simulator is used as the dynamics model, shouldn't the solution found be very similar to percentage feasibility?
>
> We have significantly expanded our metrics discussion in Section 3.1 and in Appendix Section 5.7. We have added formal definitions and long form discussion in Appendix Section 5.7. Results Section 3.1 now states: “We evaluated the performance of BELT across three metrics intended to capture its ability to plan and how robust those plans are.  "Solution Found" is the rate at which the planner returns that it has found a solution that satisfies the success criteria, capturing an algorithm's ability to explore and plan. "Success Rate" is the rate at which the solution is successfully executed if it is found (in this work we rollout successful solutions nine times), capturing how robust the plan is. This robustness may be with respect to a stochastic policy, errors in the learned model, or a stochastic environment.  "Feasibility" is the rate at which any of the replays for a given solution was able to solve the problem, capturing whether the plan was feasible (partially isolating the impact of an inconsistent policy). These are discussed in more detail in Appendix Section 5.7.”
>
> > Also, if CEM finds a solution in 3% of the cases, how is the success rate 40%? It would be really nice to explain this clearly.
>
> The success rate is only computed on rollouts when the planner returns a solution. We also note that for CEM, the high success rate is primarily due to it only solving the simplest of problems.

---

### Meta-Review · Area_Chair_xoSY · 2021-08-15

**Recommendation:** Accept (Poster)
**Confidence:** 5

**Metareview:**

### Final Meta-Review

The authors have carefully responded to reviewers' questions and, after the revision and discussion process, reviewers agree that the updated manuscript has gained in clarity. The initial concerns have been mostly resolved and all reviewers now recommend paper acceptance. I echo their recommendation—nonetheless, the authors should note the reservations expressed by reviewer FQLy and address them to the extent possible in the final version of the paper.


### Original Meta-Review

The paper presents a hierarchical planning scheme combining task-level tree search with learned motion-level control policies. The reviewers agree the the paper is generally well motivated and introduces an interesting new approach. On the other hand, the reviewers express shared concerns regarding technical clarity of terminology and definitions as well as the evaluation metrics used; raised issues regarding the comparison to the state of the art and choice of baselines also need to be addressed.

Three of the four reviewers point out the lack of an explicit upfront discussion of the three metrics used to evaluate the performance of the method (solution found, success rate, and feasibility). As far as I can tell, "feasibility" is only defined in the caption of Table 1, but its meaning (somewhat surprising) has significant implications on gauging the method's effectiveness. I ask that the authors provide clear and formal definitions of these metrics so that the reviewers can fully assess the results.

One reviewer stresses that state-of-the-art task-motion planners are able to solve problems in this class successfully, and the proposed method needs to be more strongly motivated with respect to these existing approaches. The authors must respond to these comments, also explaining why their choice of experimental baselines did not include any methods in this family.

Two of the reviewers also raise questions regarding the state space and its representation (both continuous and discrete) in the proposed algorithm. The authors should respond to these questions, clarifying what representation is being used in each case (search tree, learned policy, paper figures, etc.). Similarly, please address the questions regarding goal and task representation.

---

> ### Author Response · Authors · 2021-08-29
> **Addressed comments on Metrics, TAMP, and State representation**
>
> Thank you for your clear and constructive meta review. We have responded to each of the reviewers comments below and made particular changes based on your comments, which we believe has improved the paper. Please let us know if you have any remaining concerns or questions!
>
> > Three of the four reviewers point out the lack of an explicit upfront discussion of the three metrics used to evaluate the performance of the method (solution found, success rate, and feasibility). As far as I can tell, "feasibility" is only defined in the caption of Table 1, but its meaning (somewhat surprising) has significant implications on gauging the method's effectiveness. I ask that the authors provide clear and formal definitions of these metrics so that the reviewers can fully assess the results.
>
> We have significantly expanded our metrics discussion in Section 3.1 and in Appendix Section 5.7. We have added formal definitions and long form discussion in Appendix Section 5.7. Results Section 3.1 now states: ““We evaluated the performance of BELT across three metrics intended to capture its ability to plan and how robust those plans are.  "Solution Found" is the rate at which the planner returns that it has found a solution that satisfies the success criteria, capturing an algorithm's ability to explore and plan. "Success Rate" is the rate at which the solution is successfully executed if it is found (in this work we rollout successful solutions nine times), capturing how robust the plan is. This robustness may be with respect to a stochastic policy, errors in the learned model, or a stochastic environment.  "Feasibility" is the rate at which any of the replays for a given solution was able to solve the problem, capturing whether the plan was feasible (partially isolating the impact of an inconsistent policy). These are discussed in more detail in Appendix Section 5.7.”
>
> > One reviewer stresses that state-of-the-art task-motion planners are able to solve problems in this class successfully, and the proposed method needs to be more strongly motivated with respect to these existing approaches. The authors must respond to these comments, also explaining why their choice of experimental baselines did not include any methods in this family.
>
> We have responded to the reviewer on this and altered our related work comparison to task and motion planners. The primary difference between BELT and such approaches is that they generally consider explicit subtasks, symbols, or transition constraints, while BELT requires minimal assumptions on the task space and instead leverages the learned local policy to build the tree in a task relevant manner.
>
> > Two of the reviewers also raise questions regarding the state space and its representation (both continuous and discrete) in the proposed algorithm. The authors should respond to these questions, clarifying what representation is being used in each case (search tree, learned policy, paper figures, etc.). Similarly, please address the questions regarding goal and task representation.
>
> We have addressed these in Section 2 and revised Figure 3 such that it parallels Algorithm 1 and Section 2. We have further clarified the full state considered (which is continuous) in the Appendix 5.1 and several times in the main body of the paper as well as the task space for LMP in Section 2.

---

### Decision · Program_Chairs · 2021-09-13

**Decision:**

Accept (Poster)

**Comment:**

### Final Meta-Review

The authors have carefully responded to reviewers' questions and, after the revision and discussion process, reviewers agree that the updated manuscript has gained in clarity. The initial concerns have been mostly resolved and all reviewers now recommend paper acceptance. I echo their recommendation—nonetheless, the authors should note the reservations expressed by reviewer FQLy and address them to the extent possible in the final version of the paper.


### Original Meta-Review

The paper presents a hierarchical planning scheme combining task-level tree search with learned motion-level control policies. The reviewers agree the the paper is generally well motivated and introduces an interesting new approach. On the other hand, the reviewers express shared concerns regarding technical clarity of terminology and definitions as well as the evaluation metrics used; raised issues regarding the comparison to the state of the art and choice of baselines also need to be addressed.

Three of the four reviewers point out the lack of an explicit upfront discussion of the three metrics used to evaluate the performance of the method (solution found, success rate, and feasibility). As far as I can tell, "feasibility" is only defined in the caption of Table 1, but its meaning (somewhat surprising) has significant implications on gauging the method's effectiveness. I ask that the authors provide clear and formal definitions of these metrics so that the reviewers can fully assess the results.

One reviewer stresses that state-of-the-art task-motion planners are able to solve problems in this class successfully, and the proposed method needs to be more strongly motivated with respect to these existing approaches. The authors must respond to these comments, also explaining why their choice of experimental baselines did not include any methods in this family.

Two of the reviewers also raise questions regarding the state space and its representation (both continuous and discrete) in the proposed algorithm. The authors should respond to these questions, clarifying what representation is being used in each case (search tree, learned policy, paper figures, etc.). Similarly, please address the questions regarding goal and task representation.